# GENERALIZATION IN DATA-DRIVEN MODELS OF PRIMARY VISUAL CORTEX

**Konstantin-Klemens Lurz,**[1-2,*] **Mohammad Bashiri,**[1-2] **Konstantin Willeke,**[1-2]
**Akshay K. Jagadish,**[1] **Eric Wang,**[4-5] **Edgar Y. Walker,**[1,2,4-5]
**Santiago A. Cadena,**[2,3] **Taliah Muhammad,**[4-5] **Erick Cobos,**[4-5]
**Andreas S. Tolias,**[4-5] **Alexander S. Ecker,**[6] **Fabian H. Sinz**[1-5, **]

[1] Institute for Bioinformatics and Medical Informatics, University of Tübingen, Germany
[2] International Max Planck Research School for Intelligent Systems, Tübingen, Germany
[3] Bernstein Center for Computational Neuroscience, University of Tübingen, Germany
[4] Department for Neuroscience, Baylor College of Medicine, Houston, TX, USA
[5] Center for Neuroscience and Artificial Intelligence, Baylor College of Medicine, Houston, TX, USA
[6] Department of Computer Science / Campus Institute Data Science, University of Göttingen, Germany

[*]`konstantin.lurz@uni-tuebingen.de`, [**]`fabian.sinz@uni-tuebingen.de`

## ABSTRACT

Deep neural networks (DNN) have set new standards at predicting responses of neural populations to visual input. Most such DNNs consist of a convolutional network (core) shared across all neurons which learns a representation of neural computation in visual cortex and a neuron-specific readout that linearly combines the relevant features in this representation. The goal of this paper is to test whether such a representation is indeed generally characteristic for visual cortex, i.e. generalizes between animals of a species, and what factors contribute to obtaining such a generalizing core. To push all non-linear computations into the core where the generalizing cortical features should be learned, we devise a novel readout that reduces the number of parameters per neuron in the readout by up to two orders of magnitude compared to the previous state-of-the-art. It does so by taking advantage of retinotopy and learns a Gaussian distribution over the neuron's receptive field position. With this new readout we train our network on neural responses from mouse primary visual cortex (V1) and obtain a gain in performance of 7% compared to the previous state-of-the-art network. We then investigate whether the convolutional core indeed captures *general* cortical features by using the core in transfer learning to a different animal. When transferring a core trained on thousands of neurons from various animals and scans we exceed the performance of training directly on that animal by 12%, and outperform a commonly used VGG16 core pre-trained on imagenet by 33%. In addition, transfer learning with our data-driven core is more data-efficient than direct training, achieving the same performance with only 40% of the data. Our model with its novel readout thus sets a new state-of-the-art for neural response prediction in mouse visual cortex from natural images, generalizes between animals, and captures better characteristic cortical features than current task-driven pre-training approaches such as VGG16.

## 1 INTRODUCTION

A long lasting challenge in sensory neuroscience is to understand the computations of neurons in the visual system stimulated by natural images (Carandini et al., 2005). Important milestones towards this goal are general system identification models that can predict the response of large populations of neurons to arbitrary visual inputs. In recent years, deep neural networks have set new standards in predicting responses in the visual system (Yamins et al., 2014; Vintch et al., 2015; Antolík et al., 2016; Cadena et al., 2019a; Batty et al., 2016; Kindel et al., 2017; Klindt et al., 2017; Zhang et al., 2018; Ecker et al., 2018; Sinz et al., 2018) and the ability to yield novel response characterizations (Walker et al., 2019; Bashivan et al., 2019; Ponce et al., 2019; Kindel et al., 2019; Ukita et al., 2019).

Such a general system identification model is one way for neuroscientists to investigate the computations of the respective brain areas *in silico*. Such *in silico* experiments exhibit the possibility to study the system at a scale and level of detail that is impossible in real experiments which have to cope with limited experimental time and adaptation effects in neurons. Moreover, all parameters, connections and weights in an *in silico* model can be accessed directly, opening up the opportunity to manipulate the model or determine its detailed tuning properties using numerical optimization methods. In order for the results of such analyses performed on an *in silico* model to be reliable, however, one needs to make sure that the model does indeed replicate the responses of its biological counterpart faithfully. This work provides an important step towards obtaining such a generalizing model of mouse V1.

High performing predictive models need to account for the increasingly nonlinear response properties of neurons along the visual hierarchy. As many of the nonlinearities are currently unknown, one of the key challenges in neural system identification is to find a good set of characteristic nonlinear basis functions—so called *representations*. However, learning these complex nonlinearities from single neuron responses is difficult given limited experimental data. Two approaches have proven to be promising in the past: *Task-driven* system identification networks rely on transfer learning and use nonlinear representations pre-trained on large datasets for standard vision tasks, such as object recognition (Yamins & DiCarlo, 2016). Single neuron responses are predicted from a particular layer of a pre-trained network using a simple readout mechanism, usually an affine function followed by a static nonlinearity. *Data-driven* models share a common nonlinear representation among hundreds or thousands of neurons, and train the entire network end-to-end on stimulus response pairs from the experiment. Because the nonlinear representation is shared, it is trained via massive multi-task learning (one neuron–one task) and can be learned even from limited experimental data.

*Task-driven* networks are appealing because they only need to fit the readout mechanisms on top of a given representation and thus are data-efficient in terms of the number of stimulus-response pairs needed to achieve good predictive performance (Cadena et al., 2019a). Moreover, as their representations are obtained independently of the neural data, a good predictive performance suggests that the nonlinear features are characteristic for a particular brain area. This additionally offers the interesting normative perspective that the functional representations in deep networks and biological vision could be aligned by common computational goals (Yamins & DiCarlo, 2016; Kell et al., 2018; Kubilius et al., 2018; Nayebi et al., 2018; Sinz et al., 2019; Güçlü & van Gerven, 2014; Kriegeskorte, 2015; Khaligh-Razavi & Kriegeskorte, 2014; Kietzmann et al., 2019). In order to quantify the fit of the normative hypothesis, it is important to compare a given representation to other alternatives (Schrimpf et al., 2018; Cadena et al., 2019a). However, while representations pre-trained on ImageNet are the state-of-the-art for predicting visual cortex in primates (Cadena et al., 2019a; Yamins & DiCarlo, 2016), recent work has demonstrated that pre-training on object categorization (VGG16) yields no benefits over random initialization for mouse visual cortex (Cadena et al., 2019b). Since random representation should not be characteristic for a particular brain area and other tasks that might yield more meaningful representations have not been found yet, this raises the questions whether there are better ways to obtain a generalizing nonlinear representation for mouse visual cortex.

Here, we investigate whether such a generalizing representation can instead be obtained from *data-driven* networks. For this purpose, we develop a new data efficient readout which is designed to push non-linear computations into the core and test whether this core has learned general characteristic features of mouse visual cortex by applying the same criteria as for the task-driven approach: The ability to predict a population of unseen neurons in a new animal (transfer learning). Specifically, we make the following contributions: ❶ We introduce a novel readout mechanism that keeps the number of per-neuron parameters at a minimum and learns a bivariate Gaussian distribution for the readout position from anatomical data using retinotopy. With this readout alone, we surpass the previous state-of-the-art performance in direct training by 7%. ❷ We demonstrate that a representation pre-trained on thousands of neurons from various animals generalizes to neurons from an *unseen animal* (transfer learning). It exceeds the direct training condition by another 11%, setting the new state-of-the-art and outperforms a *task-driven* representation—trained on object recognition—by about 33%. ❸ We then show that this generalization can be attributed to the *representation* and not the readout mechanism, indicating that the data-driven core indeed captures generalizing features of cortex: A representation trained on a single experiment (4.5k examples) in combination with a readout trained on anatomically matched neurons from four experiments (17.5k examples) did not achieve this performance. ❹ Lastly, we find that transfer learning with our data-driven core is more data-efficient than direct training, achieving the same performance with only 40% of the data.

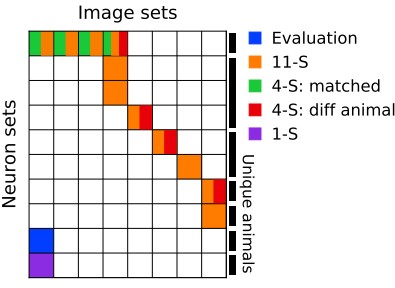

Figure 1: **Scans and training sets.** Overview of the datasets and how they are combined into different training sets. Each scan was performed on a specific set of neurons (rows) using a specific set of unique images (columns). Repeatedly presented test images were the same for all scans. Some scans were performed on the same neuron but with different image sets (first row). Colors indicate grouping of scans into training sets and match line colors in Fig. 5 to indicate which dataset a representation/core (not the readout) was trained on.

## 2 METHODS

### 2.1 DATA

**Functional data** The data used in our experiments consists of pairs of neural population responses and grayscale visual stimuli sampled and cropped from ImageNet, isotropically downsampled to $64 \times 36$ px, with a resolution of $0.53$ ppd (pixels per degree of visual angle). The neural responses were recorded from layer L2/3 of the primary visual cortex (area V1) of the mouse, using a wide field two photon microscope (Sofroniew et al., 2016). Activity was measured using the genetically encoded calcium indicator GCaMP6s. V1 was targeted based on anatomical location as verified by numerous previous experiments performing retinotopic mapping using intrinsic imaging. We selected cells based on a classifier for somata on the segmented cell masks and deconvolved their fluorescence traces (Pnevmatikakis et al., 2016). We did not filter cells according to visual responsiveness. The stimulation paradigm and data pre-processing followed the procedures described by Walker et al. (2019). A single scan contained the responses of approximately 5000–9000 neurons to up to 6000 images, of which 1000 images consist of 100 unique images which were presented 10 times each to allow for an estimate of the reliability of the neuron (see Appendix for a detailed description of the datasets). We used the repeated images for testing, and split the rest into 4500 training and 500 validation images. The neural data was preprocessed by normalizing the responses of the neurons by their standard deviation on the training set. To put the number of recorded neurons per scan into perspective, assuming that V1 has an area of about 4mm$^2$, that L2/3 is about 150-250$\mu$m thick and has a cell density of 80k excitatory cells per mm$^3$, entire V1 L2/3 should contain about 48k - 80k neurons (Garrett et al., 2014; Jurjut et al., 2017; Schüz & Palm, 1989), similar to the maximum number of neurons that we train a model on (72k neuron, 11-S , Fig. 1, orange). Note, however, that this does not mean that these 72k neurons sample V1 or the visual field of a mouse evenly because of possible experimental biases in the choice of the recording location.

All together, we used 13 scans from a total of 7 animals (Fig. 1). Each scan is defined by the set of neurons it was performed on (rows/*neuron sets* in Fig. 1) and the set of images that were shown (columns/*image sets* in Fig. 1). Different image sets had non-overlapping training/validation images, but the same test images. Some of the scans were performed on the same neurons, but with different sets of natural images (first row in Fig. 1). These neurons were matched across scans by cross-correlating the structural scan planes against functionally recorded stacks (Walker et al., 2019). Stitching data from several scans in this way allowed us to increase the number of image presentations per neuron beyond what would be possible in a single scan. We combined these scans into different training sets (one color–one training set in Fig. 1) and named each one of them—*e.g.* 11-S for a set with 11 **S**cans. The different sets are further explained in the respective experiments they are used in. All data from the seven mice used in this work has been recorded by trained personnel under a strict protocol according to the regulations of the local authorities at Balor College of Medicine.

### 2.2 NETWORKS AND TRAINING

The networks are split conceptually into two parts: a *core* and a *readout*. The core captures the nonlinear image representation and is shared among all neurons. The readout maps the features of the core into neural responses and contains all neuron specific parameters.

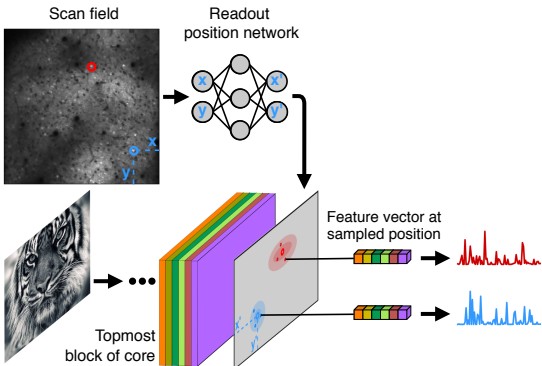

Figure 2: **Using retinotopy to learn the readout position from anatomical data.** The Gaussian readout for each neuron uses features from a single location on the final tensor of the core CNN (bottom). The position is drawn from a 2D Gaussian for every image during training. The parameters of the Gaussian for each neuron are learned during training. The means of the Gaussians are predicted from each neuron's coordinates on cortex by a *Readout Position Network* whose weights are shared across neurons and learned during training (top). During testing, the mean of the Gaussian is used as the neuron's position.

**Representation/Core** We model the core with a four-layer convolutional neural network (CNN), with 64 feature channels per layer. In each layer, the 2d-convolutional layer is followed by a batch normalization layer and an ELU nonlinearity (Ioffe & Szegedy, 2015; Clevert et al., 2015). All convolutional layers after the first one are depth-separable convolutions (Chollet, 2017) which we found to yield better results than standard convolutional layers in a search among different architecture choices.

**Readouts** We compared two different types of readouts to map the nonlinear features of the core to the response of each neuron. For each neuron, a tensor of $\mathbf{x} \in \mathbb{R}^{w \times h \times c}$ (**w**idth, **h**eight, **c**hannels) needs to be mapped to a single scalar, corresponding to the target neuron's response. All of our readouts assume that this function is affine with a linear weight tensor $\mathbf{w} \in \mathbb{R}^{w \times h \times c}$, followed by an ELU offset by one (ELU+1), to keep the response positive. Furthermore, both readouts assume that in feature space the receptive field of each neuron does not change its position across features, but they differ in how this receptive field location is constrained and learned.

The *factorized readout* (Klindt et al., 2017) factorizes the 3d readout tensor into a lower-dimensional representation by using a spatial mask matrix $u_{ij}$ and a vector of feature weights $v_k$, i.e. $w_{ijk} = u_{ij}v_k$. The spatial mask $u_{ij}$ is restricted to be positive and encouraged to be sparse through an L1 regularizer.

Our novel *Gaussian readout* reduces the number of per-neuron parameters. It computes a linear combination of the feature activations at a single spatial position— parametrized as $(x, y)$ coordinates —via bilinear interpolation (Sinz et al., 2018). To facilitate gradient flow during training, we replace the spatial downsampling used in (Sinz et al., 2018) by a sampling step, which during training draws the readout position of each $n^{\text{th}}$ neuron from a bivariate Gaussian distribution $\mathcal{N}(\mu_n, \Sigma_n)$ for each image in a batch separately. This is the sampling version of (St-Yves & Naselaris, 2017) where the readout location is weighted spatially with a Gaussian profile. In our case, $\mu_n$ and $\Sigma_n$ are learned via the reparametrization trick (Kingma & Welling, 2014). Initializing $\Sigma_n$ large enough ensures that there is gradient information available to learn $\mu_n$ reliably. During training, $\Sigma_n$ shrinks as the estimate of the neuron position improves. During evaluation we always use the position defined by $\mu_n$, making the readout deterministic. This version of the Gaussian readout has $c + 7$ parameters per neuron (2 for $\mu$, 4 for $\Sigma$ because the linear mapping in the reparametrization trick is $2 \times 2$, and 1 for the scalar bias).

The second innovation of our Gaussian readout is to couple the location estimation of single neurons by exploiting the retinotopic organization of primary visual cortex (V1) and other areas. Since V1 preserves the topology of visual space, we estimate a neuron's receptive field location from its position $\mathbf{p}_n \in \mathbb{R}^2$ along the cortical surface available from the experiments. To that end, we learn a common function $\mu_n = f(\mathbf{p}_n)$ represented by a neural network that is shared across all neurons (Fig. 2). Since we work with neurons from local patches of V1, we model $f$ as a linear fully connected network. This approach turns the problem of estimating each neuron's receptive field location from limited data into estimating a single linear transformation shared by all neurons, and reduces the number of per-neuron parameters to $c + 5$. We initialized the *Readout Position Network* to a random orthonormal 2-2 matrix scaled by a factor which was optimized in hyper-parameter selection.

Finally, when training on several scans of anatomically matched neurons from the same mouse (see Data), we share the feature weights $v_k$ across scans. To account for differences in spike inference between scans, we introduced a scan-specific scale and bias for each neuron after the linear readout. We mention in the respective sections whether features are shared or not. The bias of each readout is initialized with the average response on the training set. The effects of both feature sharing and learning from cortical anatomy on the performance of the readout are shown in the Appendix.

**Training**    The networks were trained to minimize Poisson loss $\frac{1}{m} \sum_{i=1}^{m} \left( \hat{r}^{(i)} - r^{(i)} \log \hat{r}^{(i)} \right)$ where $m$ denotes the number of neurons, $\hat{r}$ the predicted neuronal response and $r$ the observed response. We used early stopping on the correlation between predicted and measured neuronal responses on the validation set (Prechelt, 1998): if the correlation failed to increase during any 5 consecutive passes through the entire training set (epochs), we stopped the training and restored the model to the best performing model over the course of training. We found that this combination of Poisson objective and early stopping on correlation yielded the best results. After the first stop, we decreased the learning rate from $5 \times 10^{-3}$ twice by a decay factor of $0.3$, and resumed training until it was stopped again. Network parameters were iteratively optimized via stochastic gradient descent using the Adam optimizer (Kingma & Ba, 2015) with a batch size of 64. Once training completed, the trained network was evaluated on the validation set to yield the score used for hyper-parameter selection. The hyper-parameters were then selected with a Bayesian search (Snoek et al., 2012) of 100 trials and subsequently kept fixed throughout all experiments. Only the scale of the readout regularization was fine-tuned with additional Bayesian searches for the cases of different amounts of data independently. In transfer experiments, we froze all parameters of the core and trained a new readout only.

**Evaluation**    We report performance as *fraction oracle* (see Walker et al., 2019), which is defined as the correlation of the predicted response and the observed single-trial test responses relative to the maximally achievable correlation measured from repeated presentations. We estimated the oracle correlation using a jackknife estimator (correlation of leave-one-out mean against single trial). Per data point, we trained 25 networks for all combinations of five different model initializations and five random partitions of the neurons into core and transfer sets. The image subsets were drawn randomly once and kept fixed across all experiments except in Fig. 5 where the full neuron set was used and 5 random partitions of image subsets were drawn instead. We selected the best performing models across initializations and calculated 95% confidence intervals over neuron- or image seeds.

## 3    RESULTS

We investigated the conditions under which a *data-driven* core generalizes to new neurons in the same or different animals. We did this by pre-training a core on differently composed datasets (core sets, see 3.1) and testing that core in transfer learning to a new set of neurons (transfer set, see 3.1). Our main finding is that transferring a core trained on multiple scans (up to 35k unique images and 70k unique neurons) to a new set of 5335 unique neurons from a single scan (4.5k unique images) yields an improvement in performance of about 12% compared to a network directly trained on the single scan. This result was independent of whether the transferred core was trained on the same animal or not. By carefully choosing the Transfer learning conditions, we can attribute this boost in performance to the generalization of the core.

### 3.1    TRANSFER LEARNING CONDITIONS

To test the generalization performance of the core, we used a separate set of 1000 neurons which we call *transfer set*. These neurons were not used to train the transferred core but only to fine-tune a new readout (note that Fig. 4 and 5 use different transfer sets, for intra- and inter- animal transfer respectively). A transfer set is a subset of neurons, not images. Thus, each transfer set also had a train, validation, and test split of its images. We compared the performance in transfer learning to that of a network directly trained end-to-end on the transfer set or subsets thereof (*direct* condition). Because the core of the *direct* condition was not transferred, it could adapt to the neurons at hand giving it a fair chance to outperform the transferred cores.

In the transfer learning conditions (all except *direct*), the core was always trained on a separate set of neurons (*core set*) and subsequently frozen, while the readout was always trained on top of

the frozen core using the transfer set. Again, a core set is a subset of the neurons, not images. In order to quantify the generalization of the core, we needed to decouple the amount of data it takes to train the core from the amount of data it takes to train the readout. We thus considered two transfer conditions: *diff-core/best-readout* and *best-core/diff-readout*. In the *best-core/diff-readout* condition, the core was trained on the core set using all images, while the readout was trained on the transfer set using different numbers of images. This condition tests the data efficiency of the readout. We expect that better cores lead to a higher data efficiency in the readout, *i.e.* require less data to achieve good performance. In the *diff-core/best-readout* condition, we trained the core on the core set using different numbers of images while the readout was trained on the transfer set using all images. Thereby we tested how the generalization of the core is affected by the amount of data used to train it.

## 3.2 DIRECT TRAINING AND WITHIN ANIMAL/ACROSS NEURON GENERALIZATION

The following transfer experiments (Fig. 3 and 4) aim at investigating how the number of images and neurons provided to the core and the readout affects generalization to new neurons. To this end, we used a dataset that includes as many images as possible while still providing a reasonably large number of neurons. The `4-S:matched` dataset (first row in Fig. 1, green) provides 17596 images and 4597 neurons anatomically matched across 4 scans. Each scan was performed on the same neurons, but showing different sets of images (test set was identical). The dataset was concatenated along the image dimension and split into core and transfer sets of 3597 and 1000 neurons.

**Direct training** We used the `4-S:matched` core set to compare the performance of the Gaussian and factorized readout in the *direct* condition on the original core set before transfer. We tested both readouts with and without feature sharing (see Networks and training). The performance of the networks increased with the number of images (Fig. 3). It also increased with the number of neurons, but the number of images had a far stronger effect. While the performance saturated quickly with the number of neurons, saturation w.r.t. the number of images did not seem to be reached, even when using all 17.5k images. The Gaussian readout outperformed the factorized readout in predictive performance by 7% fraction oracle for the full `4-S:matched` dataset, reaching $0.886 \pm 0.005$ and $0.826 \pm 0.005$ fraction oracle respectively (mean±std). While the Gaussian readout profits from feature sharing, the factorized readout is hurt by it (Fig. 3, light vs. dark colors). This might be because the spatial masks in the factorized readout are less constrained in contrast to the Gaussian readout where the position network and the usage of only a single readout point exerts a stronger inductive bias. In all future experiments, we thus use feature sharing only for the Gaussian readout.

**Within animal/across neuron generalization** For both readouts, the generalization performance of the learned core, tested in the *diff-core/best-readout* condition, increased with the number of images used to train the core (Fig. 4, pink). The cores and readouts were trained on the core and transfer set of the `4-S:matched` dataset. As before, the Gaussian readout outperformed the factorized readout, exhibiting a stronger increase in performance with the number of images and a better final performance when the entire dataset was used to train the core. Even for a core trained on few data, a readout can yield good performance if it has access to enough images (pink line). Importantly,

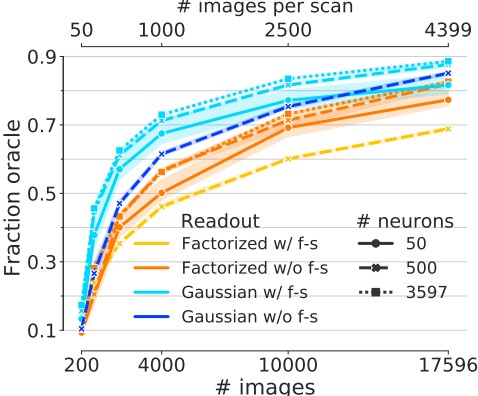

Figure 3: **Performance of end-to-end trained networks.** Performance for different subsets of neurons (linestyle) and number of training examples (x-axis). The same core architecture was trained for two different readouts with and without feature sharing (color) on the matched neurons of the `4-S:matched` core set (Fig. 1, green). Both networks show increasing performance with number of images. However, the network with the Gaussian readout achieves a higher final performance (light blue vs. orange). While the Gaussian readout profits from feature sharing (light vs. dark blue), the factorized readout is hurt by it (yellow vs. orange). Shaded areas depict 95% confidence intervals across random picks of the neuron subsets.

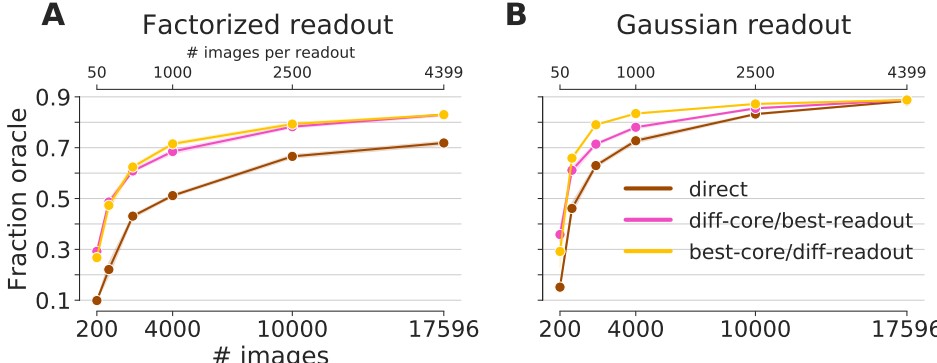

Figure 4: **Generalization to other neurons in the same animal.** A core trained on 3597 neurons and up to 17.5k images generalizes to new neurons (pink and yellow line). A core trained on the full data yields very good predictive performance even when the readout is trained on far less data (yellow). If the readout is trained with all data, even a core trained on few data can yield a good performance (pink). Both transfer conditions outperform a network directly trained end-to-end on the transfer dataset (brown). For the full dataset, all training conditions converge to the same performance. Except in the *best-core/diff-readout* condition for very few training data, the Gaussian readout (**B**) outperforms the factorized readout (**A**). The data for both the training and transfer comes from the 4-S:matched dataset (Fig 1, green). Not that the different number of images can be from the core or transfer set, depending on the transfer condition.

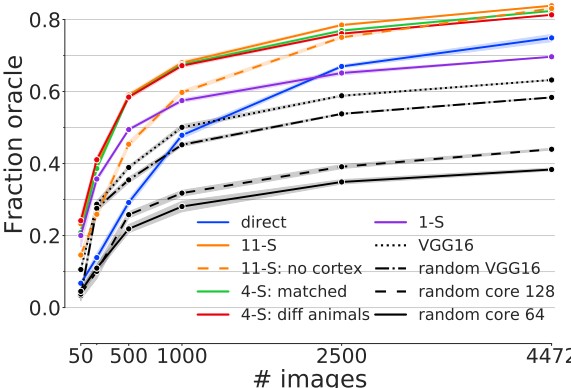

Figure 5: **Generalization across animals.** Prediction performance in fraction oracle correlation as a function of training examples in the transfer set for a Gaussian readout (x-axis) and different ways to obtain the core (colors). The transfer training was performed on the evaluation dataset (blue, Fig 1). Cores trained on several scans used in transfer learning outperform direct training on the transfer dataset (blue line; direct condition).

except for very low numbers of images, the fastest increase in performance occurred in the *best-core/diff-readout* condition, where a core trained on all images (but different neurons) was frozen and a new readout was trained on the transfer set for varying numbers of images (yellow lines). This result shows that a data-driven core provides general characteristic features for mouse V1, and these features generalize to new neurons. Importantly, for 4k images (about the size of a typical experiment), the performance of the *best-core/diff-readout* was approximately 7% better than the performance of the *diff-core/best-readout* condition ($0.834 \pm 0.003$ and $0.780 \pm 0.007$ fraction oracle respectively). This observation that a readout on all 17.5k images on a core from 4k images could not reach the performance of a readout on 4k images on a core from 17.5k images suggests that the better performance is due to the generalizing core and not the readout.

## 3.3 GENERALIZATION ACROSS ANIMALS

So far, we tested generalization performance of data-driven cores to different neurons in the same animal using the 4-S:matched dataset. A stronger test for generalization is transfer learning across animals, where we may have to deal with inter-subject variability. To this end, we compared several cores derived from different core sets, random initialization, or ImageNet pre-training in terms of their generalization to neurons from a mouse which was not used to train the core. The transfer set consisted of 5335 neurons from a different mouse presented with images that were also in the

core set (Fig. 1, blue). Note that performance was still evaluated on a set of test images that were neither used to train core nor readout. Apart from the previously used core set with anatomically matched neurons (`4-S:matched`, Fig. 1, green), we also trained a core on a single scan from another animal (`1-S`, Fig. 1, purple), and on a set of four scans from different animals with different images (`4-S: diff animals`, Fig. 1, red). Finally we also trained a core on all datasets together, without any information on neuron matching or image sets (`11-S`, Fig. 1, orange). For completeness, we also compared to a task-driven VGG core (Simonyan & Zisserman, 2015), a randomly initialized VGG core (reading out from conv2-2 for both) (Cadena et al., 2019b), our core (64 features) with randomly initialized weights, and a scaled up version of our core with the same number of features as the VGG (128). As before, we compared the generalization performance to a model trained under the *direct* condition trained on the transfer set. Due to its better performance, we used the Gaussian readout for these experiments but a version with the factorized readout can be found in the Appendix.

We tested the generalization in the *best-core/diff-readout* condition on a single transfer set (Fig. 5), *i.e.* all performances are reported on the same transfer set (5335 neurons) (blue, Fig. 1 and 5), but differ in where the core was trained (colors, Fig. 5). The most striking finding is that representations trained on several scans not only reached better performances for fewer training images, but they also reached a better overall performance for the available data from a single scan compared to *direct* training (Fig. 5 orange, green, and red vs. blue). Interestingly, this improvement did not require matched neurons in the core set (Fig. 5) and any potential negative effects of inter-subject variability were outweighed by the benefits of using multiple scans to train the core. The two cores trained on a single scan reached about the same final performance (Fig. 5, blue vs. purple), with a slightly better performance for the directly trained model, as expected. However, the final performance of both single scan models was about 10% smaller than the transfer performance from the models pre-trained on four and eleven scans, respectively (Fig. 5, blue and purple vs. red, green, and orange). We provide visualizations of the receptive fields – produced via response maximization (Walker et al., 2019; Bashivan et al., 2019) – of some example neurons obtained from our best model (Fig. 5, orange at 4472 images) in the Appendix, Fig. 4.

Consistent with previous work (Cadena et al., 2019b), pre-trained and random VGG16 cores performed similarly (Fig. 5 gray, dotted vs. dash-dotted). Both VGG cores performed worse than a directly trained *data-driven* core (Fig. 5 blue vs. gray dotted and dash-dotted). Our core with random weights (64 features) performs worst, demonstrating that training on neural data extracts characteristic features. Scaling up this random core to VGG size (128) does not match its performance which could be due to the interaction of the initialization with architectural differences.

Lastly, we investigated the effect of constraining all neurons to share the transformation from cortical location to receptive field location by temporarily deactivating this feature (Fig. 5, `11-S: no cortex` dashed orange), and found that this constraint was particularly useful for small numbers of images. An equivalent figure for Fig. 5 for the factorized readout, a table with a detailed overview over the most important results in numerical form, as well as a comparison with other performance metrics can be found in the Appendix.

# 4 RELATED WORK

The idea of a common cortical feature representation is wide-spread in sensory and systems neuroscience, going back to the idea of V1 as a bank of Gabor filters or edge detectors (Jones et al., 1987; Olshausen & Field, 1996). A substantial body of recent work focuses on feature representations learned by training deep networks on vision tasks such as object recognition (Cadena et al., 2019a; Yamins & DiCarlo, 2016; Güçlü & van Gerven, 2014; Kriegeskorte, 2015; Khaligh-Razavi & Kriegeskorte, 2014). The brain-score[1] initiative compares different representations, resulting from pre-training on different tasks or different network architectures, with regards to performance of multiple neural prediction tasks (Schrimpf et al., 2018). In contrast to that, we focused on whether multi-task learning between thousands of neurons leads to a generalizing representation.

While *task-driven* representations perform comparably to *data-driven* representations in primates (Cadena et al., 2019a), Cadena et al. (2019b) recently demonstrated that they show almost no difference in predictive performance for mice. Our results corroborate this finding (Fig. 5) and show that a

---

[1] https://www.brain-score.org/

*data-driven* representation outperforms a *task-driven* representation by a substantial margin, even when tested on equal grounds with transfer learning.

Sharing a representation between neurons is commonly used to learn *data-driven* system identification networks (Cadena et al., 2019b;a; Batty et al., 2016; Sinz et al., 2018; Antolík et al., 2016; Klindt et al., 2017). Klindt et al. (2017) investigated the effect of the number of neurons and training examples onto the predictive performance of a data-driven network. However, this experiment was done on simulated data only and did not explore the generalization (transfer) performance of the learned representation. To the best of our knowledge, we are the first to systematically investigate the ability of data-driven representations to capture general characteristic features of visual cortex.

## 5 DISCUSSION

Machine learning applications in biology are often faced with limited amount of data. Especially, for recent deep learning approaches this poses a challenging problem. One promising way to approach it is multi-task learning by training a shared nonlinear representation on multiple tasks or subjects. This increases the data volume and can help to extract inductive biases to achieve better generalization. Here, we investigated a particular instance of this problem: Modeling the responses of thousands of cortical neurons as a function of natural visual stimuli. We demonstrated that nonlinear *data-driven* representations, trained via massive multi-task learning through parameter sharing among thousands of neurons from mouse primary visual cortex, generalize to other neurons and mice, and significantly outperform common *task-driven* alternatives that are predictive for monkey V1 (Cadena et al., 2019b; Yamins & DiCarlo, 2016). As noted by Cadena et al. (2019b) already, this does not imply that all task-trained representations are necessarily suboptimal but rather indicates that we have not found the right task yet. But so far, our network sets a new state-of-the-art for neural response prediction in direct training *as well as* transfer learning.

Our transfer results strongly suggests that data trained cores can capture features that are characteristic of mouse primary visual cortex, and the fact that the Gaussian readout can predict novel neurons from this core with relatively few training example corroborates this idea. In addition, the good transfer learning performance indicates that inter-subject variability, which could affect the success of multi-task learning, does not seem to be a major problem for this application. For that reason, we believe that our *data-driven* core is the most characteristic representation to date to predict mouse primary visual cortex and that it could be a great tool in new experiments where data is scarce and/or training time is limited, such as the *inception loops* introduced by Walker et al. (2019) and Bashivan et al. (2019). To facilitate this, we share the weights of the trained representation together with its code online[2] to allow others to predict neural responses with it. Other possible applications include for example the analysis of the learned feature representations to investigate the operations in visual processing, or using the core to support vision tasks like image categorization (Li et al., 2019). Additionally, we also share the dataset that we evaluate our core on (Fig. 1, blue) so that other representations can be tested and compared with ours on the same data[3].

Our results are based on "deconvolved" calcium traces of neural activity integrated over 500 ms. Whether data driven cores can also provide generalizing features for shorter time scales, recordings with electrodes, to dynamic responses, or of higher areas remains to be seen in future studies.

### ACKNOWLEDGMENTS

We thank A.K. Schalkamp, A. Nix, C. Blessing, S. Safarani, E. Froudarakis, and N. Patel for comments and discussions. KKL is funded by the German Federal Ministry of Education and Research through the Tübingen AI Center (FKZ: 01IS18039A). FHS is supported by the Carl-Zeiss-Stiftung and acknowledges the support of the DFG Cluster of Excellence "Machine Learning – New Perspectives for Science", EXC 2064/1, project number 390727645. SAC and ASE are supported by the German Research Foundation (DFG grant EC-479/1-1). This work was supported by an AWS Machine Learning research award to FHS. MB was supported by the International Max Planck Research School for Intelligent Systems. Supported by the Intelligence Advanced Research Projects Activity via Department of Interior/Interior Business Center contract number D16PC00003.

---

[2]https://github.com/sinzlab/Lurz_2020_code
[3]https://gin.g-node.org/cajal/Lurz2020

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
