# OpenReview forum: "Generalization in data-driven models of primary visual cortex"
_ICLR.cc/2021/Conference — ICLR 2021 Spotlight_

### Official Review · AnonReviewer4 · 2020-10-28
**Interesting findings**

**Rating:** 7
**Confidence:** 3

**Review:**

The paper presents an experimental study on predicting the responses of mice V1 neurons with computational models. The paper advances a few contributions:

1. Confirm that task-driven models based on object recognition, are outperformed by data-driven models for predicting single neuron responses.
2. Show that training a shared model of neural responses on data from several animals and several neurons leads to models that transfer well to data from new neurons and new animals.
3. Introduce a novel readout mechanism that allows models to be shared fully across neurons which in turn helps transfer learning.

I think this paper is interesting and it should be presented at ICLR. I am not an expert in this specific sub-field so I am not qualified to make suggestions or evaluate the experimental design. I will leave here a few suggestions that I hope you will consider for a camera ready version.

1. Is the claim that, in mice, task-driven models are outperformed by data-driven models fully justified? I have no trouble believing that object recognition is not the right task for mice, but there is a fair chance we just haven't found a good task yet, and that we might find it in the future. If this is correct, it might be worth mentioning in the introduction or discussion.

2. I think the Introduction could do a better job of anticipating the implications of this study, similarly, the discussion is a bit dry and does little more than just repeating the results. As I mentioned, I am not especially familiar with the literature in this particular subfield, so I had a hard time imagining what I should learn about visual systems in general from your study. What can we do with this new information?

3. Would it be possible to visualize the receptive fields you learn? Maybe some unit maximization technique could be sufficient. I think it would be cool to see a few.

4. It might make sense to move the training regimes for direct training, within- and across-animal transfer to the method section and explain the data splitting, training and evaluation after those have been introduced. The reader will know why you are designing your datasets and training a certain way, which might make it easier to follow.

Thank you again for sharing these cool ideas and results, I hope my suggestions help.
All the best!

---

### Official Review · AnonReviewer2 · 2020-10-28
**Review for "Generalization in data-driven models of primary visual cortex"**

**Rating:** 6
**Confidence:** 5

**Review:**

The authors adopt a data-driven approach to neural system identification. They train a neural network consisting of a "core" and a "readout" in an end-to-end fashion to learn stimulus (visual inputs) -- response (single neuron activity) pairs. Since the core is shared across neurons, these stimulus-response pairs can be learnt in a massively parallel manner. In particular, they propose a novel readout mechanism that is parameter efficient and drives the core to learn better and generalizable features of the visual inputs. They find that their representations are more suited to predict neural responses in the mouse visual cortex when compared to representations derived from task-driven learning, especially in the context of transfer to previously unseen animals. Lastly, they also observe that the combination of their core+readout is more sample efficient than other naive alternatives.

Pros:
One of the major positives about this paper is the presented dataset. It seems to be relatively large and well-curated. This can certaily support several follow-up studies.

The authors identify that "global" use of features (i.e. the full hXwXc representational tensor) in the readout is a wasteful strategy (in terms of learned parameters per neuron) and instead adopt a local approach where they only select specific feature columns per neuron to drive the readout. Though this is of minor technical novelty, this constraint forces the core to learn appropriate representations while allowing the entire module to be more data efficient, given the big reduction in the number of free parameters.

The sample efficiency studies are neat and informative. The dissociations gleaned from diff-core/best-readout vs best-core/diff-readout scenarios are useful. Though it needs more work to fully justify this claim, their demonstration that transferred representations seem to be more effective than direct training is surprising and interesting.

Cons:
Though this study is certainly valuable, the manuscript needs several clarifications before it can be publication-ready.

(i) The authors seek to develop better core representations indirectly by controlling the readout mechanism. This is fine, but there is little justification as to why they chose the current "core" architecture. This choice contains arbitary decisions (such as including depth-separable convolutions) that are not justified. Was there a systematic procedure behind a search that led to this architecture? Were other non-standard architecures tested?

(ii) Figure 2 currently seems to be adding very little value and needs to be improved. Given that the proposed readout mechanism was a major contribution of this paper, the authors could have used the Fig. 2 space to visually depict this readout procedure, on top of the readout position network. The arrow to a neuron is also a bit misleading.

(iii) One of my main concerns is with respect to the liberal use of the term "generalization". The authors repeatedly state that train-val-test splits were based on neurons and not images. This, coupled with the fact that their readouts leverage retinotopy, it is surprising that the authors never discuss the spatial segregation of the "held-out" neurons (say H) from the neurons in the training set (say T). If most H neurons were spatially proximate to T neurons, then this amounts to an "interpolation" regime for the network as opposed to "extrapolation". If my understanding here is wrong, could the authors please clarify why?

(iv) The authors report that transfered "core" representations work better than direct-training in their generalization experiments. This result is surprising and needs to be more strongly justified computationally. Is it possible that a sub-optimal training regime was used for direct-training? Is this anyhow related to issue (iii) raised above?

(v) The authors report that task-driven cores (such as VGG-16 pretrained on imagenet) perform badly in generalizaing across animals. Is this due to impoverished data regimes? Or are there more systemic issues? Also, VGG-16 isn't the best ventral stream model that best fits neural data. Do the authors think that this claim would hold for more recent task-driven systems, like CORnet-S for example.

(vi) Though not necessary for this manuscript per se, it would be helpful to test the usefulness of the generalizable core representations presented here for visual tasks supported by early visual areas. Perhaps some commentary on this would be nice.

Minor:
"Code for the analyses and the weights of the best generalizing representation will be shared in the final version of the paper". The authors do not commit to making the dataset public. Is this oversight or intentional?

how sensitive to number neurons in a scan?

Clarity: (Fig. 4 caption) "a fully trained core": I think the authors are referring to a core trained on all available data, which is different from "fully training" a network as this alludes to loss saturation. Also language like "a sub-optimal" core is vague and misleading.

---

> ### Author Response · Authors · 2020-11-12
> **Clarifications for points (iii) and (iv)**
>
> Thank you very much for your detailed feedback! We are positive that we can address all your points in the rebuttal. We just briefly wanted to clarify a few items regarding your points (iii) and (iv). We feel there might have been a misunderstanding and we want to make sure we understand your point correctly so we can address it adequately. We would kindly ask you to comment whether this addresses your points or clarify your concerns for us:
>
> Regarding (iii):
>  - We are not sure whether we succeeded at describing the train-, val- and test-splits properly. While it is true that we split our neurons into a train set (for the core) and a test set, our final results that we report are always evaluated on a separated test-set of images. Specifically, our training scheme consists of the three following steps: 1) train core & readout on neuron set T and training image set; 2) freeze the core, train a new readout on neuron set H and training image set; 3) report score in neurons H and test image set. By that we test how well the core generalizes/transfers to novel neurons
>
> - The information we get from retinotopy concerns solely the readout positions of the neurons, not their feature weights. The feature weights of different neurons are still learned separately even if the neurons are spatially close in the cortex. In that sense, the readout feature weights are not interpolated.
>
> - Finally, our main figure (Fig. 5) shows generalization across animals. This means that neurons from set T and H are from two different animals and could not possibly suffer from any effect of spatial proximity.
>
> Regarding (iv):
> - The results showing that a transfer core can outperform direct-training are very promising indeed. Please note though that while the direct-training regime has access to the test neurons themselves (again, the final performance is always reported on a separate test set of images), the core from the transfer regime has access to a greater number of neurons (from 11 datasets in the orange line, Fig. 5) and images (up to 4 different training stimulus sets). As you pointed out correctly, direct training should outperform transfer training if the data size is comparable, which is exactly what we find (Fig 5, purple line). You can also see that performance of direct training is not saturated yet. If it were possible to record more responses from the same neurons, we would expect direct training to reach the performance of the transfer regime eventually.

---

> > ### Comment · AnonReviewer2 · 2020-11-13
> > **Are neurons here truly I.I.D?**
> >
> > Thanks for the clarification. I do better understand the train/test protocol better now. Off the bat, I do agree that transfer across animals is a good demonstration. Perhaps my question can be paraphrased. Your transfer argument (within the same animal) to a held-out set of neurons points towards "more neurons in training" leads to better performance on "neurons in test set". It is unclear to me as to what is the knowledge that is being transferred if it does not support feature weight interpolation (directly or indirectly). If two neurons are spatially proximate, and one responds to an input in a particular manner, then you can reasonably guess that the other one would respond in a similar manner. This was the rationale behind my original comment. The neurons are clearly not independently and identically distributed as is the case for typical train/test examples in ML applications. Clarifying how you think this dependency structure is exploited by your system can help.
> >
> > Re: (iv) Thanks for outlining this. The significance of blue (1-S) and purple (direct) being close to each other is pretty easy to miss in Figure 5. Perhaps the authors should make this more obvious.

---

> > > ### Author Response · Authors · 2020-11-16
> > > **Generalization between neurons**
> > >
> > > Response
> > > Thanks for your quick response. Re (iv): We’ll bring out this detail to be more salient in the next iteration of this figure.
> > >
> > > Regarding your other comment: The transfer between animals clearly shows that it cannot be due to spatial interpolation among similarly tuned neurons in close spatial proximity, since it’s still there even when the neurons are from two different animals. In addition, in contrast to orientation columns in primates, features of cortical neurons in mice are locally more diverse (sometimes referred to as “salt and pepper”). Even if that were the case though, our network does not provide a mechanism that allows two neurons in the same scan to share information about the feature weight vector in the readout.
> > >
> > > The reason why we think that training on many neurons helps is the same reason why transfer learning works with imagenet. Pretraining the convolutional part with many neurons helps to identify good basis functions (representations) such that learning a readout on similar problems becomes more data efficient/feasible. An illustration of this idea would be the following: assume that the entirety of  V1 would be made up of simple cells with Gabor filters. Since we use a convolutional core, a feature learned at one spatial position is available on all other spatial positions. This enables the model to learn shared features among neurons regardless of their receptive field position. So with enough neurons, we would eventually identify the basis set of all relevant Gabors (a kind of dictionary if you want). At this point, learning new ones should be easier.
> > >
> > > This reasoning is consistent with figure 4, where the best core (convolutional part) is trained on 3597 neurons and 17596 images. Following the logic above, we think that this identifies a good set of convolutional features (core), such that merely learning a new readout for 1000 neurons on top of the frozen core (yellow curve) is more data efficient. Thus, the learning curves rise quicker compared to directly training the entire network on 1000 neurons (brown curve).

---

> > ### Comment · Area_Chair1 · 2020-11-13
> > **availability of the dataset**
> >
> > Hi there -- R2 lists the dataset as one of the major positives about this paper. I glanced quickly at the paper and could not find any clear statement about releasing the data. Could you please clarify whether the data will be released or not? Thanks

---

> > > ### Author Response · Authors · 2020-11-16
> > > **Availability of datasets and model weights**
> > >
> > > Thanks for your comment. The networks in our manuscript are trained on 13 two photon scans recorded by our experimental co-authors. As you note, the datasets are very valuable, scientifically and because they take a lot of effort and funding to record. Since this manuscript was not the main reason why these datasets were recorded, we cannot share all datasets (yet), to give the experimental scientists who recorded them enough time to finish their analyses and publish their work which will include a large two-photon data release.
> > >
> > > However, we also see the need for reproducibility. For this reason we want to find a middle ground between these interests. We will share the evaluation dataset (one scan, used to test in figure 5). We actually have made that set public already (but not shared the location in the paper yet). We will make a note of this and the online location in the final version (to not break anonymity). In addition, we will make the network trained on 11 scans (fig 5) publicly available. With that, people can reproduce the key parts of figure 5 and parts of the data are available for further research.

---

> ### Author Response · Authors · 2020-11-19
> **Changes to the paper addressing your suggestions**
>
> We have now uploaded a new version of the paper containing the following changes according to your suggestions:
>
> 1. In order to come up with the core architecture, we had done extensive searches among architecture choices both in this as well as in other projects. We now added this information to the section “Networks and Training”.
> 2. Thanks for this feedback. We have now adjusted Figure 2 to contain the whole readout procedure, not just the readout position network. We also replaced the arrow to the neuron with a stylized spike train.
> 3. Please refer to the discussion in the previous responses to you.
> 4. Please refer to the discussion in the previous responses to you.
> 5. As noted by Cadena and colleagues 2019, whose results are consistent with ours, one reason why task driven networks like VGG16 trained on imagenet perform poorly on mouse V1 prediction could be that image classification might not be the right task for mouse visual cortex. There might be tasks which are better suited for this purpose in the future (see also Reviewer 4) but until then our network provides a predictive alternative. We do not think that the poor performance of image-classification task-driven approaches is due to our choice of network (VGG16). VGG16 actually outperforms CORnet-S on V1 (0.294 for CORnet-S vs. 0.355 for VGG16 on brain-score.org) Comparing our data-driven network against a large number of pre-trained architectures would be interesting, but is beyond the scope of the current work. We changed our discussion to make clear that we do not want to imply that all task-driven representation must necessarily be suboptimal in mouse visual cortex.
> 6. We added a comment and a reference on this in the discussion section.
>
> Minor: Concerning the publication of the datasets and model, please refer to our response to the Area Chair.
>
> Clarity: We re-phrased the respective paragraphs.

---

### Official Review · AnonReviewer3 · 2020-10-30
**Innovative state-of-the-art predictor of mouse visual responses**

**Rating:** 8
**Confidence:** 4

**Review:**

Summarize what the paper claims to contribute.
The paper introduces a deep-network-based approach to regression of responses to natural stimuli in mouse primary visual cortex. There is closely related work in the literature, but this paper achieves very good performance, partly through a new way of accounting for neurons’ receptive-field positions. The paper also provides a helpful analysis of prediction performance versus numbers of images and neurons used to train the model, and shows that the already excellent performance is not saturated with respect to the number of images. The work also shows that features learned by a core network generalize well across different mice.

List strong and weak points of the paper.
Strong points:
-	Empirical modelling of neural responses has a long tradition, and the results in this paper are state-of-the-art.
-	Thorough and insightful positioning in the recent literature.
-	Expert execution in terms of details of the technical work.
-	The method of parameterizing the receptive field location is well-motivated and effective.
-	The analyses are interesting and provide useful insights.

Weak points:
I wouldn’t characterize any part of the paper as weak, but here are some minor suggestions to further strengthen it:
-	Say more about how the model can be used, or what insights might arise from it (there is only a short comment on inception loops).
-	Say more about limitations as a model of neural responses, particularly with respect to dynamics. While the method is impressive with respect to short-time-window responses, system identification methods have long been used to study temporal responses. I think a short comment on this scope limitation would help to further contextualize the paper.
-	An additional way to contextualize the results might be relative to the total number of neurons in L2/3 of VISp (I believe ~200K). Does this number have any significance relative to the dimension of the core-network output, or the number of recorded neurons?
-	Consider adding a sentence on ethics oversight regarding the animal experiments.
-	Consider adding a few further details of the experiments.

Clearly state your recommendation (accept or reject) with one or two key reasons for this choice.
I recommend that the paper be accepted. The paper addresses a long-standing problem very well. It introduces a new method that is well justified and effective. Overall, the performance is impressive, and the analyses are well done.

Ask questions you would like answered by the authors to help you clarify your understanding of the paper and provide the additional evidence you need to be confident in your assessment.
What are the kernel sizes in the core?
Which hyperparameters are adjusted in the hyperparameter search?

Provide additional feedback with the aim to improve the paper.
I was confused by the following sentences:
“… both readouts assume that the receptive field of each neuron is the same across features”
“… readout has c + 7 parameters per neuron …” (I only see c+6.)
“Fig 5 for the factorized readout …” (I didn’t get it until reading it four times and looking for these results in Figure 5 twice.)

---

### Official Review · AnonReviewer1 · 2020-11-02
**Predicting V1 responses with less training data**

**Rating:** 8
**Confidence:** 4

**Review:**

The authors train a neural net to predict responses of mouse V1 L2/3 neurons to visual stimulation. The NN has a "core" that is shared between all neurons, and a neuron-specific readout. They train the core on multiple animals and find that it can generalize well: it can be used in a new animal and (with sufficient training of the readouts) achieve high performance. They also use a neat approach of constraining the readout weights (receptive field location) using the known retinotopy of V1. Finally, they show that their network outperforms task-trained ones at predicting V1 responses.

This is nice work overall. I have a few suggestions:

1) It might be worth considering other measures of performance, different from the normalized correlation coefficient. Recent work shows that this measure can have unintended bias, being substantially affected by trial-to-trial variability.
See "The unbiased estimation of the fraction of variance explained by a model" from Pospisal and Bair (https://doi.org/10.1101/2020.10.30.361253) for details, and a potential solution.

2) 2-photon imaging can have issues at detecting single spikes (see this preprint, for example: Relationship between spiking activity and simultaneously recorded fluorescence signals in transgenic mice expressing GCaMP6,  https://doi.org/10.1101/788802). So the neural dataset could in principle show more multi-spike events than single-spike ones, or have other issues. This is inevitable of course with calcium imaging, but it makes the problematic to compare with previous work that used electrical recordings. E.g., I don't think it is possible to prove better performance for this work than the prior ones, because of this difference in recording methods. A good follow-up work should try this method on electrical recordings from (say) monkey, and compare with performance from the Cadena, Yamins, Kindel, Klindt, Batty, etc. studies.

---

> ### Author Response · Authors · 2020-11-23
> **Changes to the paper addressing your suggestions**
>
> Thank you for your positive feedback and suggestions on how to further improve it. We agree with you on both your issues:
>
> 1. Thank you for pointing us towards this interesting paper for an unbiased estimation of FVE. We included an equivalent figure to our Fig. 5 with this measure in the appendix of the updated paper. In short: our data passes the threshold for a stable computation of the unbiased FVE and the relative ordering of all the curves are preserved, corroborating our results. Note though, that the calculation of the unbiased FVE assumes that the data is variance stabilized. However, in order to properly do this, we would have needed to retrain the models on the variance stabilized data which we had not done during the experiments and could not add during the constrained time of this rebuttal period. For the sake of the comparison, however, we still added the comparison with this metric for our most important results to the appendix.
>
> 2. We are aware of the shortcomings of calcium imaging and it might be that multi-spike events which are detected as single spikes could pose a problem. Note, however that we use “deconvolved” calcium signals. Also note that we predict  the responses over a large time window of 500ms. Even if a multi-spike even is detected as a single large spike, the effect of that might be cancelled out by this integration.
> While electrical recordings are more precise in the temporal dimension, they currently come at the expense of much fewer neurons being recorded simultaneously. Our method takes advantage of having the recordings of many neurons by sharing the computations in the core network between these neurons. It thus benefits from the ability of calcium imaging to yield a massive amount of neurons. For a different project, we are currently investigating whether transferring data-driven cores can also work for electrophysiological recordings of less neurons, but the results were not finished at the time of the rebuttal...

---

### Decision · Program_Chairs · 2021-01-07
**Final Decision**

**Decision:**

Accept (Spotlight)

**Comment:**

This paper has received four positive reviews. The main intellectual contribution of the paper is the introduction of a novel readout mechanism that allows models to be shared fully across neurons which in turn helps transfer learning across neurons and even across animals. The reviewers commented on the technical strength of the paper. At the same time, the main contribution remains relatively incremental from a technical standpoint, and while the approach may be of value to future work, the impact of the current study on neuroscience (which is the target here) is quite limited. Nonetheless, there seems to be sufficient enthusiasm from the reviewers to recommend this paper be accepted.